# QBB: Quantization with Binary Bases for LLMs

**Adrian Bulat**[1,2]    **Yassine Ouali**[1]    **Georgios Tzimiropoulos**[1,3]
[1]Samsung AI Cambridge    [2]Technical University of Iasi    [3]Queen Mary University of London

## Abstract

Current post-training quantization methods for LLMs compress the weights down to 4-bits, with moderate to low degradation in accuracy. However, further reducing the number of bits or accelerating the network while avoiding large accuracy drops, especially for smaller, sub 7B models, remains an actively researched and open problem. To address this, in this work, we introduce Quantization with Binary Bases (QBB), a new approach for low-bit quantization that effectively removes (nearly) all multiplications, reducing the implementation to summations. Our novel approach works by decomposing the original weights into a set of binary (1-bit) matrices using an iterative process. For a given layer, starting from a weight matrix, we first construct an initial approximation using an analytical solution, where each new binary matrix, paired with a scaling vector, approximates the residual error of the previous estimation. Secondly, using gradient descent and a progressive learning curriculum, we find the optimal set of binary matrices and scaling vectors that minimize the $\ell_2$ distance between the produced approximation and original weights. Thirdly, as previous steps are input agnostic, we holistically optimize the scaling vectors alone, calibrating them in student-teacher fashion, with the teacher providing both the data, by autoregressive generation starting from a random token, and the target logits. When evaluated across multiple LLM families, our approach matches and outperforms all prior works, setting a new state-of-the-art result using a summation-only based approach.

## 1 Introduction

Large Language Models (LLMs) demonstrated proficiency in natural language understanding and generation across multiple domains, exhibiting strong in-context learning abilities [3; 52; 53]. This remarkable performance can be in part attributed to the ever-increasing model and dataset sizes [25; 9], with the current generation of models being trained on Trillion-sized (tokens) datasets [53], using Billions of parameters [53; 24; 29]. This, in turn, made deploying such models, even on consumer-grade GPUs, problematic. A promising and flexible direction that disentangles the training and the deployment process, while being hardware friendly, is given by post-training and/or data-free quantization [16; 32; 48]. Within this area, current methods have successfully achieved $4\times$ compression (i.e. 4-bit quantization) with small drops in accuracy [16; 7; 32], especially for larger (7B+) models. However, compressing the model further and/or applying said methods to smaller models yields unsatisfactory results [16; 32]. Importantly, on the inference side, this class of PTQ approaches replaces 16b multiplications with mixed-precision (i.e., 16b-4b) ones, which have limited hardware support and rely on software-based solutions. Our aim is to alleviate these limitations, enabling higher compression rates while removing (nearly) all multiplications.

To this end, departing from prior work, we focus on the most extreme case of quantization, that of binarization (which, as we will show, enables us to remove costly multiplications). As binarizing the weights directly, without retraining, results in unsatisfactory performance, we propose to approximate the weights $\mathbf{W}$ using a set of binary matrices $\mathbf{B}_i, i \in \{1, ..., N\}$ with corresponding fp16 scaling vectors $\boldsymbol{\alpha}_i$. The main case of interest studied is that of $N = 4$. We note that binarising the weights

is a particularly interesting case as the matrix multiplications can be implemented using masked selection/data loading and summations, removing all the multiplications, but 1, that with the scaling vector.

As binarising the weights directly is suboptimal (see Sec. 4), we introduce a new iterative approach for estimating the optimal binary matrices $\mathbf{B}_i$ and scaling vectors $\boldsymbol{\alpha}_i$: Firstly, for a given weight $\mathbf{W}$ belonging to the $l$-th layer, we construct an initial approximation using an analytical solution, whereby each new additional binary matrix approximates the residual error of the previous estimation(s). Secondly, using gradient descent, we iteratively adjust the binary weights and scaling vectors one by one by minimizing the $\ell_2$ norm between the target weights and the produced estimation. As until now the process is agnostic to the input distribution of the data, we plug the approximated weights back into the model, optimizing in the interest of speed and generalizability the scaling vectors only. This step uses a teacher-student approach and does not require any training dataset. Instead, we generate synthetic data starting from random tokens by adapting [38] to be more sample efficient. Note that this bypasses the need for any real data. Our approach matches and outperforms prior 4-bit quantization methods across a wide variety of LLMs models (e.g: LLaMA-2, Phi-2, etc.) while removing all multiplication (except the final one with the scaling vector), being fast to calibrate and being amenable, in certain conditions to higher compression rates (see Sec. 4.6), setting a new state-of-the-art result.

## 2   Related work

**LLMs quantization** is a particular application of network quantization [18; 8; 22; 42; 56] to LLMs. As for the former, broadly speaking, these methods can be split into two categories: Quantization-Aware Training (QAT) - which requires retraining or training in quantized form, and Post-Training Quantization (PTQ), which can be applied directly, without additional retraining. As retraining such models, especially without losing generality, is challenging, most of the work focused on PTQ [16; 49; 32; 30], although a recent line of work tackles the latter too using data-free strategies [38] or parameters efficient fine-tuning [14]. One of the most prominent recent lines of work was the introduction of GPTQ [16] that proposes a one-shot low-bit weight quantization method based on approximate second-order information. Follow-up work improves upon it by considering the outliers [27; 13], the effect of the activations on the weight quantization performance [32; 30], by training the quantization parameters [48] or by grouping and reordering the parameters [62]. Methodologically, our approach stands in between PQT and QAT, as we support both layer-wise input agnostic binarization and fast data-free recalibration. In either case, our approach doesn't require a training corpus.

**Binarization** also known as 1-bit quantization, represents the most extreme case of quantization. In this instance, when both operands are binary, the multiplications can be replaced with bitwise operations, while when one is binary and the other real, the multiplications become summations. These desirable properties make binarization a particularly interesting case of quantization. The current base formulation can be traced to [10; 11], which proposes to binarize the values using the `sign` function, whose gradient is estimated using a straight-through estimator [2]. Subsequent works primarily focus on training-aware solutions for vision models [46; 33; 34; 5; 4; 15; 58; 64; 63; 26; 6; 40; 35; 61; 45]. For example, Rastegari et al. [46] introduces analytically computed scaling vectors for both the input and the weights, while [4] building on top, proposes to learn them holistically, via backpropagation instead. Liu et al. [34] argues for the use of real-valued down-sampling layers and the use of double skip connections, while [5] replaces ReLU with PReLUs and makes use of two-staged optimization.

Many works have also studied network binarization in the context of transformers [1; 44; 36; 37], focusing on either full retraining or fine-tuning on labeled datasets using mostly smaller models (e.g., BERT, BART). More relevant to our work are the recent attempts to adapt and apply binarization to LLMs [55; 39; 47; 60]. BitNet [55] swaps all linear layers with BitLinear, their 1-bit weight's replacement, which are then trained from scratch. PB-LLM [47] explores the partial binarization of LLMs, in which the salient weights are stored in higher bits, and the rest are binarized, with a fine-tuning step (i.e., a QAT step) to recover the original LLM performance. BitNet b1.58 [39] encodes every weight using three states $\{-1, 0, 1\}$ instead of two, which are then trained from scratch similar to BitNet [55]. Finally, OneBit [60] proposes a 1-bit QAT method that decomposes the weights into 1-bit weights and two full-precision scaling vectors, which are then fine-tuned with a

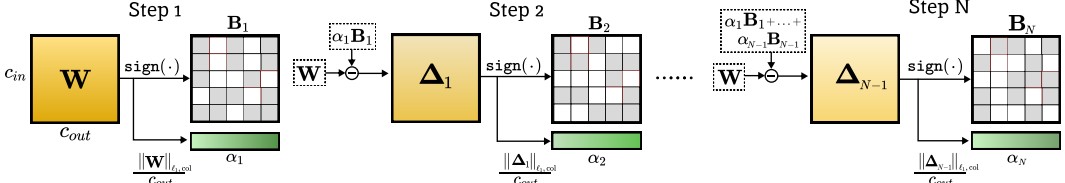

Figure 1: **Binary weights initialization phase**: starting from the target weight $\mathbf{W}$, we construct $N$ binary matrices $\mathbf{B}_i$ and scaling vectors $\boldsymbol{\alpha}_i$ analytically. The 1st binary weight and scaling vector are obtained using Eq. 2 while the subsequent ones use Eq. 3.

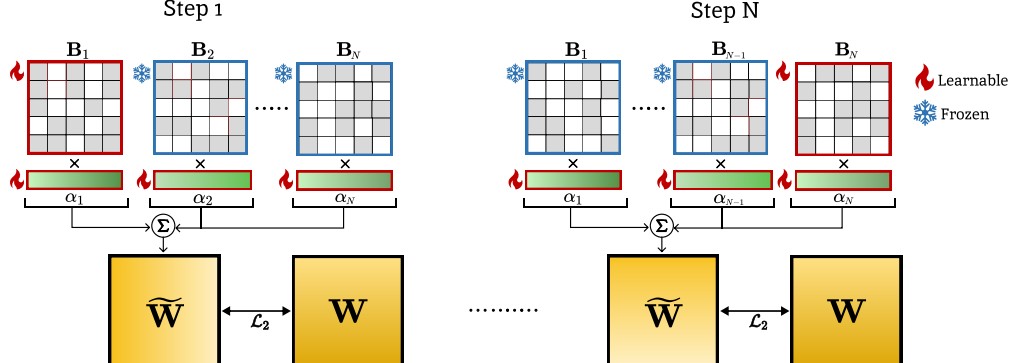

Figure 2: **Iterative weights binarization:** Starting from the weights initialized as shown in Fig. 1, we update the binary weights one-by-one, where at each step $i \in \{1 \dots N\}$, only $\mathbf{B}_i$ is updated by minimizing the loss function in eq. (4), while the rest are frozen. Note that all of the scaling vectors $\boldsymbol{\alpha}_i$ are updated at every step.

distillation loss for a better approximation. However, all these approaches require full retraining or fine-tuning on labeled datasets with a large training overhead. In contrast, our approach is applicable post-training, doesn't require any labeled (nor unlabeled) datasets, is fast to optimize, makes use of multiple binary bases, and scales well to billion-sized models.

## 3 Method

### 3.1 Layer-wise input-agnostic weights binarization

Given a linear layer, represented as $f(\mathbf{X}, \mathbf{W}, \mathbf{b}) = \mathbf{X} \cdot \mathbf{W} + \mathbf{b}$, with $\mathbf{X} \in \mathbb{R}^{n \times c_{in}}$ the input tensor, $\mathbf{W} \in \mathbb{R}^{c_{in} \times c_{out}}$ the weight matrix and, optionally, $\mathbf{b} \in \mathbb{R}^{c_{out}}$ the bias term. $n$ represents the number of tokens, $c_{in}$ and $c_{out}$ the number of input, and respectively output channels. Note that we are considering the batch size to be 1 and the bias 0 for brevity, as the formulations are independent of them. Our goal is to then quantize the elements of $\mathbf{W}$ with $\widetilde{\mathbf{W}}$ such that $\mathbf{X} \cdot \mathbf{W} \approx \mathbf{X} \cdot \widetilde{\mathbf{W}}$. Unlike prior works [16; 32] that focus primarily on 3 and 4-bit quantization, herein, we focus on compressing the weights using a set of binary bases, without comprising the model's accuracy and importantly, without requiring any training or calibration dataset(s), enabling in the process a potential implementation based (nearly) solely on summations.

For our purposes, let us consider the most extreme case of quantization, that of binarization, whereby the values are represented using 1-bit. In this instance, the real-valued weights $\mathbf{W}$ can be approximated as $\mathbf{W} \approx \widetilde{\mathbf{W}} = \boldsymbol{\alpha}\mathbf{B}$, where $\mathbf{B} = \texttt{sign}(\mathbf{W})$ is a binary matrix and $\boldsymbol{\alpha} \in \mathbb{R}_{+}^{c_{out}}$ a scaling vector. Given a set of real-valued weights that are binarized, following Rastegari et al. [46], the optimal value for $\boldsymbol{\alpha}$ can be computed analytically as $\boldsymbol{\alpha} = \frac{1}{c_{out}} \|\boldsymbol{W}\|_{\ell_1, \text{col}}$ with $\|\boldsymbol{W}\|_{\ell_1, \text{col}}$ as the $\ell_1$-norm computed over the columns of $\boldsymbol{W}$. Note, that in order to avoid spurious states (i.e. states at which the function is ill-defined - 0), the $\texttt{sign}$ function used is defined as: $\texttt{sign}(x) = \begin{cases} -1, & x < 0 \\ 1, & x \geq 0 \end{cases}$.

In essence, this relation tells us that we can find the optimal values for $\mathbf{B}$ and $\boldsymbol{\alpha}$ by solely looking at $\mathbf{W}$, i.e. without requiring any training data. The downside is that the quantization errors are very high due to reduced representation power, which completely degrades the model's performance. This is perhaps not surprising, as the number of representable states for one given value drops from $2^{16}$ for 16-bits to $2^4$ for 4 bits and, finally, $2^1$ for the binary case. This discrepancy is even more noticeable when considering the total number of unique permutations representable on such a weight matrix.

To address this, we propose to represent $\widetilde{\mathbf{W}}$ with a linear combination of N binary matrices:

$$\widetilde{\mathbf{W}} = \sum_{i=1}^{N} \boldsymbol{\alpha}_i \mathbf{B}_i \tag{1}$$

In this case, finding the optimal values for $\boldsymbol{\alpha}_i, \mathbf{B}_i, i \in \{1, ..., N\}$ can no longer be achieved using the analytical solution of Rastegari et al. [46]. As such, we propose a new iterative optimization approach in which the weights and the scaling vectors are iteratively learned via gradient descent. The exact optimization process requires, however, careful considerations. The gradients of `sign(.)` are 0 almost everywhere, taking the form of the Dirac Delta function. Hence, to make training possible, in practice they are approximated using a straight-through estimator [2] taking the form of a clipped identity function [46; 11]. These gradients are however noisy with the training prone to oscillations and instabilities: for a value infinitesimally close to 0, there exists a full state transition to either of the extremes, i.e. $-1$ or $+1$, for any non-zero gradient, hence any noise, even a machine representation error, can result in large swings. To alleviate this, we study (i) how to initialize the weights and the scales, and (ii) how to stabilize the optimization process.

**Weights and scale initialization:** Given an initial estimation $\boldsymbol{\alpha}_1 \mathbf{B}_1$ computed analytically by directly approximating the weight matrix $\mathbf{W}$ as:

$$\mathbf{B}_1 = \texttt{sign}(\mathbf{W}), \boldsymbol{\alpha}_1 = \frac{1}{c_{out}} \|\boldsymbol{W}\|_{\ell_1, \text{col}}, \tag{2}$$

we then initialize the subsequent $\boldsymbol{\alpha}_i, \mathbf{B}_i, i \in \{2 \dots N\}$ values by approximating the residual error $\boldsymbol{\Delta}_i$ at step $i$:

$$\mathbf{B}_i = \texttt{sign}(\boldsymbol{\Delta}_i), \boldsymbol{\alpha}_i = \frac{1}{c_{out}} \|\boldsymbol{\Delta}_i\|_{\ell_1, \text{col}}, \quad \boldsymbol{\Delta}_i = \mathbf{W} - \sum_{j=1}^{i-1} \boldsymbol{\alpha}_j \mathbf{B}_j. \tag{3}$$

This sets the underlying distribution behind each $\mathbf{B}_i$ matrix to the direction that would minimize the overall quantization error. See Fig. 1 for an illustration.

**Iterative weights and scales optimization:** Starting from the above initialization, we aim to minimize the following loss function:

$$\mathcal{L}_2 = \|\mathbf{W} - \sum_{i=1}^{N} \boldsymbol{\alpha}_i \mathbf{B}_i\|_2^2, \tag{4}$$

where both $\boldsymbol{\alpha}_i$ and $\mathbf{B}_i$ are trainable parameters, with $\mathbf{B}_i$ maintained in the binary state using the sign function. However, the direct, *naïve* approach of optimizing all parameters jointly leads to training instability. To alleviate this, we devise an iterative training procedure, akin to block coordinate descent, in which the weights are trained one-by-one. Specifically, and as shown in Fig. 2, for $N$ binary matrices, we have $N$ training steps. During each training step, only one binary weight is updated, while the rest stay frozen. We note, however, that the scales are trained in all cases. This approach prevents us from concomitantly changing too many binary weights, effectively simplifying the problem at each step to learning a binary correction factor to an existing approximation.

**Progressive weights quantization:** To further simplify the approximation of full-precision weights using binary bases, we follow previous binarization approaches [5; 40] and progressively reduce the gap between full-precision and binary weights. Specifically, instead of directly approximating full-precision weights, we first quantize them to a lower precision using an off-the-shelf method (e.g., 4-bit using GPTQ). Then, we apply the proposed method with these quantized weights as a starting point (i.e. initialization followed by iterative optimization).

## 3.2 Data-free holistic binarization

Until now, our binarization process was input-agnostic, considering only the to-be-binarized weights $\mathbf{W}$ on a per-layer basis. However, this does not factor two important aspects: (1) the effect of cumulative errors resulted from imperfect approximations of the previous layer's weights and (2) the effect on the overall model's output objectives, given that a local approximation isn't necessarily the optimal global solution under quantization induced errors.

To maintain the generalization properties of the target LLM, instead of making use of downstream datasets, we generate synthetic data using the (full-precision) model itself, following Liu et al. [38]. Specifically, the 1st token is randomly selected, with the model generating the subsequent ones up to a certain length or until the EOS token is encountered. Note that the next token is selected stochastically to increase the diversity of the output sequences.

Once the data is generated, we can directly use it to train the model with the standard next-token CE loss. However, as the synthetic data is noisy, forcing these labels into the quantized model is suboptimal. Instead, we directly use the produced logits $p^{\mathcal{T}}$ as soft ground truth, effectively performing Knowledge Distillation [20]. In practice, we use the MSE loss between the student (quantized model) and teacher (full precision model) logits, which is analogous to a cross-entropy based distillation at high-temperature [20]. The loss is defined as:

$$\mathcal{L}_{\texttt{MSE}} = \sum_{v=1}^{V} \sum_{i=1}^{n} \|p_{i,v}^{\mathcal{T}} - p_{i,v}^{\mathcal{S}}\|_2^2, \tag{5}$$

where $V$ represents the size of the vocabulary and $n$ the number of tokens.

By inspecting the per-layer quantization (Sec. 3.1), we notice that different layers within different regions of the model have different quantization errors that are sometimes different by up to an order of magnitude. Hence, training them only with a signal applied at the very end of the model is suboptimal. Thus, we also apply a layer-wise MSE loss between the corresponding features of the teacher and student and the end of every transformer block:

$$\mathcal{L}_{\texttt{feat}} = \sum_{l=1}^{L} \sum_{i=1}^{n} \|f_{i,l}^{\mathcal{T}} - f_{i,l}^{\mathcal{S}}\|_2^2, \tag{6}$$

where $f_{i,l}^{\mathcal{T}}$ and $f_{i,l}^{\mathcal{S}}$ are the features after the $l$-th transformer module of the $i$-th token produced by the teacher, and respectively student (quantized) model.

The overall loss is then computed as:

$$\mathcal{L}_{\texttt{distill}} = s_1 \mathcal{L}_{\texttt{MSE}} + s_2 \mathcal{L}_{\texttt{feat}}, \tag{7}$$

with $s_1$ and $s_2$ scaling factors for balancing the two loss terms.

It's generally accepted that higher accuracy gaps between the student-teacher pair hinder the learning process, as the student has a difficult job of matching the teacher's output. To alleviate this, herein, we randomly swap parts of the teacher model with blocks from the student model. Effectively, this creates an instant family of models, with a performance situated between that of student and teacher models. The sampling is varied, with increasing difficulty (i.e. fewer blocks swapped) as the training progresses.

**Efficient Calibration:** To improve the efficiency of our calibration process, firstly, we only fine-tune the scaling vectors $\boldsymbol{\alpha}_i$ making the training process compute and time efficient, keeping everything else, including the binary weights $\mathbf{B}_i$, frozen. Secondly, to speed up the training further and make it more sample-efficient, we propose a simple filtering step of the generated training data. Specifically, we only train on self-generated sequences where the quantized student and full-precision teacher differ the most, instead of directly using all the generated samples as in Liu et al. [38]. This is done by first scoring the generated sequences using Eq. 5, then keeping the top-$K$ samples with the highest student-teacher discrepancy. This reduces the training time without any impact on the final results.

## 4 Ablation studies and analysis

### 4.1 Impact of the proposed components

Table 1: **Impact of the main components** measured for a Phi-2 (2.7B) and LLaMA-2 (7B) model in terms of perplexity on WikiText2. N/A - indicates failure ($> 10^5$ perplexity).

| Method | Phi-2 (2.7B) | LLaMA-2 (7B) |
|---|---|---|
| without Sec. 3.1 & 3.2 | N/A | N/A |
| + Sec. 3.1 | 12.32 | 5.37 |
| + Sec. 3.2 | 10.48 | 5.21 |

In this section, we showcase the importance of each proposed component, focusing in particular on what it takes to make the binarization stable. For a breakdown of the KD subcomponents, see Sec. 4.3. As the results from Tab. 1 show, both proposed components, i.e. Layer-wise input agnostic weights binarization (Sec. 3.1) and Data-free holistic distillation (Sec. 3.2) improve the model's accuracy and are critical for achieving convergence. Notably, the data-free calibration is particularly important for smaller models, which are generally harder to quantize.

Moreover, to validate that binarization is inherently unstable, we first ablate the initialization step. In Fig. 3 we plot, on a layer-by-layer basis, the $\ell_2$ quantization error for different weight initialization strategies: random initialization - the

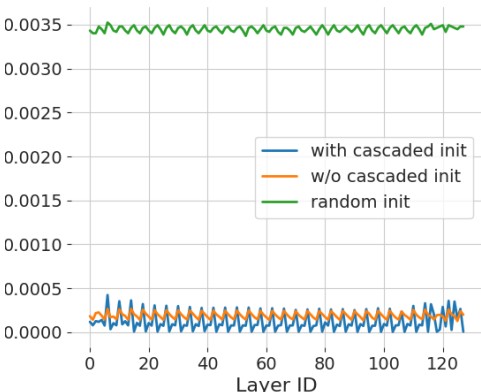

Figure 3: Per-layer reconstruction error for a Phi-2 model when varying the initialization used: random, without the cascaded (residual) init, and with.

weights are sampled from a normal distribution; w/o cascaded initialization - all weights attempt to minimize the global error at initialization; with cascaded initialization - each weight approximates the remaining residual error (Eqs. 2 and 3). We can immediately observe that random initialization results in errors that are an order of magnitude higher. Furthermore, the proposed cascaded initialization results in reconstruction errors that are on average $2\times$ smaller than the direct alternative, that minimizes the global error instead of the residual one, as proposed in our work. Interestingly, there are a few layers (the ones harder to quantize) for which the two are behaving similarly. Such layers tend to have more outliers, which makes the finding of the optimal solution (that minimizes the average reconstruction error), more prone to local minimums.

## 4.2 Binary weights and process analysis

To offer some insights on the binarization process, in Fig. 4 we show the proportion of the weights that change state between initialization and the end of the iterative training process, on a per-layer basis, for each binary base. Interestingly, the first and last binary matrix undergo minimal changes, with typically only 0.01 - 0.2% of the weights transitioning to a different state. In comparison, the second, and particularly the third base change approximately 40% and respectively 50% of their bases. Moreover, we can observe that for layer IDs 6 and 8, while the proportion of transitions decreases for the second one, the amount increases for the third, which appears to compensate in order to maintain the same overall proportion of bit flips when aggregated across all bases. These phenomena can be tied to how the initialization is performed: the first binary matrix directly approximates the

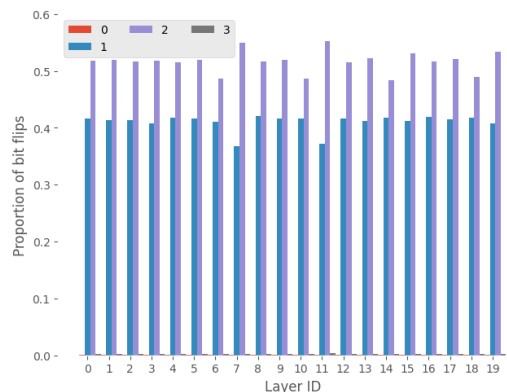

Figure 4: Proportion of binary weights that changed their state (bit flips) after applying the proposed iterative training process, shown for the first 20 layers of a Phi-2 model for all 4 bases.

full precision weights $\mathbf{W}$, hence it is already close to its optimal solution while the last one has to approximates a relatively small residual error, as 3 bases give relative similar results (see Fig. 6).

## 4.3 Effect of distillation

Knowledge Distillation [20] was shown to improve the performance of smaller models by the virtue of providing guidance either at the feature level [19] or at the output space, by providing (soft) pseudo-labels. Herein, we ablate various design choices, summarizing the results in Tab. 2. In particular, we consider the following options: (1) CE - using directly the synthetic data as hard labels and then calibrating the model using the standard next token prediction Cross-Entropy Loss; (2) KD - Using the soft probabilities provided by the teacher, this is identical to standard Knowledge Distillation [20]; (3) KD-MSE - Compute a $\ell_2$ (MSE) loss between the logits of the student and teacher; (4) KD-MSE with swap - gradually vary the strength of teacher, from weaker to stronger, by randomly swapping some block in the teacher model with ones from the student and then apply the KD-MSE loss. Note that the per-layer feature loss (eq. 6) is used for all cases except (1) CE, i.e. when the teacher model is not utilised. As the results show, the KD-MSE with swap outperforms all other variants. Moreover, for this use case, the KD-MSE variant outperforms the vanilla KD loss.

Table 2: **Effect of different Knowledge Distillation** strategies measured using a Phi-2 (2.7B) model in terms of perplexity on WikiText2.

| CE | KD | KD-MSE | KD-MSE with swap |
|---|---|---|---|
| 11.82 | 10.61 | 10.48 | **10.40** |

## 4.4 Effect of data filtering

During the calibration step, in order to make the training more sample efficient, we proposed to filter the randomly self-generated data and only keep the sequences with the highest teacher-student discrepancy as measured by the MSE between their output logits (i.e. Eq. 5). As shown in Fig. 5, this simple filtering step results in faster convergence for the same amount of training samples (i.e. lower perplexity) compared to directly training on the randomly generated sequences.

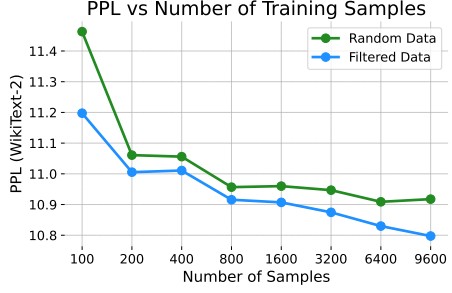

Figure 5: The PPL on WikiText-2 with Phi-2 (2.7b) trained on a variable number of training samples. Here, we compare the performance when using the generated data directly, or when filtering is applied.

## 4.5 How many binary bases should we use?

Throughout this work, we mainly focus on the $N = 4$ case as it provides an optimal balance between accuracy, compute cost and storage. For clarity and completeness, herein, we explore how the performance varies for different values of $N$. As a study case, we take the Phi-2 (2.7B) model and plot the initial reconstruction errors, after applying the method described in Sec. 3.1. We do so on a layer-by-layer basis, as this will also reveal wherever certain layers are harder to quantize. Looking at the results shown in Fig. 6 we can make the following observations: (1) There is a notable performance drop and phase transition for $N \leq 2$; (2) Perhaps unsurprisingly, not all layers are made equal, with different layers having different approximation errors, and hence exhibiting variable difficulty levels from

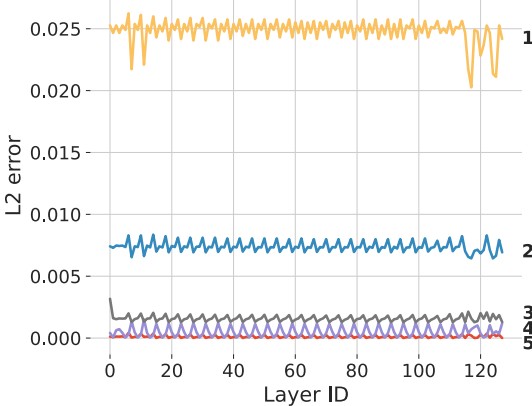

Figure 6: Per-layer reconstruction error after the initial layer-by-layer optimization for a Phi-2 model. Notice that the performance is generally stable for $N \geq 3$.

a quantization point of view; This could be indicative for future quantization efforts; (3) For the layers located in the middle, a pattern emerges, with every block being (nearly) equally hard/easy to quantize. This aligns well with the idea that transformers learn a similar representation across most of their layers, unlike, let's say, a convolutional model; (4) For $N = 5$, we can observe improvements, especially for the most difficult layers, while the easier one has a similar value as for $N = 4$.

### 4.5.1 Effect of iterative optimisation

Herein, we measure the impact of the proposed iterative training strategy presented in Sec. 3.1. As the results from Fig. 7 updating all weights jointly leads to signigicantly worse results, leading to an average error that is one order of magnitude higher.

### 4.5.2 Effect of progressive weights quantization

In line with prior work on binarization [40], we aim to gradually increase the quantization level, as this simplifies the problem by reducing the apparent difficulty of the target. To this end, in Tab. 3 we measure the impact of progressive quantization and showcase that our approach is amenable to different quantization targets. On one hand, we can observe that attempting to directly quantize the fp16 weights results in de-

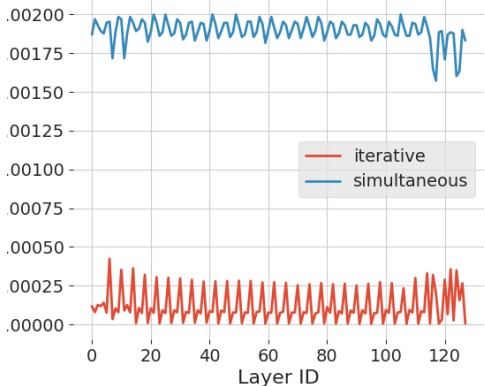

Figure 7: Per-layer reconstruction error after the initial layer-by-layer optimization for a Phi-2 model. Notice that the performance is generally stable for $N \geq 3$.

graded performance, on the other, our approach works with target weights produced by different approaches, in this case: QPTQ and OmniQuant. Moreover, we can observe that better int4 weights also translate in better binary bases, with our method incorporating the improved scale and outlier handling from OmniQuant for example.

Table 3: **Effect of progressive weight quantization** strategies measured using a LLama-2 (7B) model in terms of perplexity on WikiText2.

|  | FP16 | GPTQ | OmniQuant |
| --- | --- | --- | --- |
| Starting perf: | 5.12 | 5.61 | 5.58 |
| Ours: | 7.10 | 5.49 | 5.21 |

### 4.6 Discussion on efficiency

Current low-bit post-training quantization methods result in implementations that replace the full/half precision multiplications present within a linear layer with ones performed using operators represented on fewer bits, most frequently 16b-4b (activation-weights). When the number of bits for at least one of the operands drops to the most extreme case, i.e., 1-bit, all these multiplications are removed. Depending on the implementation, they become either a masked selection ($\{0, 1\}$) or a conditional sign set ($\{-1, 1\}$). Herein, we opt in for the former, as depending on the target hardware, it is possible to reduce the number of summations too, by implementing a masked load, given that the binary matrices generally have a sparsity level between 50-80%. Getting back to the core message, by removing all the multiplications inside the linear layer, except for the final one with the scaling vector, the power consumption could be reduced, and on custom hardware, it could translate into a smaller die size and faster execution [51; 50; 31; 57], as in principle an adder circuit can be implemented using a flip-flop. While different implementation and technological nodes will result in some variations, taking as an example [21], on a 45nm node, and depending on the bit-width of the activation, an adder consumes $4 - 8\times$ less energy compared with a multiplier. While left for future studies, further decreasing the activations too to 1-bit can enable even faster implementations based solely on bit-wise operations [4; 10]. Memory-wise, as we learn a sum of binary bases, certain implementations, although not trivial to construct, can exploit either a left-most significant bit alignment to compress

the weights to $\log_2(N+1)$ bits, as a sum of $N$ binary matrices results in $N+1$ states, or can make use of codebooks given the cross-matrix similarities and sparsity levels. All in all, we highlight that such representation, which removes (nearly) all the multiplications, opens the door to important energy savings, which are especially important in a world where the total energy consumption of AI models is expected to reach 85-135 TWh by 2027 [12] (if current trends follow) - this equates to the entire power consumption of the Netherlands.

## 5 Results

We compare our approach with the current state-of-the-art for low-bit quantization in terms of perplexity score on the main benchmark for quantization - WikiText2 [41], focusing mainly on the LLaMA-2 [53] {7, 13, 70}B family of models. However, we also include results for LLaMA [52] {7, 13, 30, 65}B and Phi-2 [23] 2.7B models. This allows us to effectively cover a wide range of model architectures and sizes, from the "smaller", and hence harder to quantize, Phi-2 (2.7B parameters) model to the large 70B LLaMA-2 model.

**Training details.** We implement our method using PyTorch [43], sourcing the pretrained models from Hugging Face [59] repo, as provided by their respective authors. The evaluation code is based on [17]. During the input-agnostic quantization part, presented in Sec. 3.1 and using a single A100 GPU, we optimize each set of binary matrices and scaling vectors, layer by layer, using the following hyperparameters: Adam optimizer [28], 15000 iterations, no weight decay, an initial learning rate of $1e-4$ decayed to $0$ using a cosine scheduler. For the data-free distillation step, presented in Sec. 3.2, we fine-tune the scaling vectors only for 2 epochs using an Adam optimizer, a cosine learning rate scheduler, no weights decay, and an initial learning rate set to $2.5e-4$. For added stability, we clip the gradients with a norm higher than 1.

**Baselines.** For weight-only quantization, we compare with the primary baseline which is the vanilla Round-to-nearest (RTN) quantization, in addition to current state-of-the-art methods for both W4A16 (i.e. weights in 4-bits and activations in 16-bits) and W3A16 quantization. Specifically, we consider GPTQ [16], SpQR [13], QuIP# [54], and OmniQuant [48]. Here, we report the results for both per-channel weight quantization and group-wise weight quantization with a group size of 128 (i.e. denoted as g128).

**Outcome.** As the results from Tab. 4 show, our approach matches or outperforms prior PTQ approaches across all model sizes.

## Broader impact and limitations

As our work removes nearly all multiplications, it could improve the energy efficiency of the current generation of AI models, reducing costs while being more environmentally friendly.

In terms of limitations, our work builds on top of existing pre-trained LLMs, some trained on closed datasets, hence any potential biases present in the original data, affecting the original models would likely propagate to the quantized models too. Therefore, we recommend caution before deploying such models. Moreover, the benefits and gains mentioned are theoretical, as no custom implementation was constructed. Additional effort may be required to translate the theoretical gains into practice.

## 6 Conclusions

In this work, we introduced Quantization with Binary Bases (QBB), a new approach for low-bit quantization that effectively removes (nearly) all multiplications, reducing the implementation to summations. Our new approach decomposes the original weights into a set of binary matrices and scaling vector(s) using an iterative process, whereby starting from an initial approximation obtained using an analytical solution, each new binary matrix and scaling vector is optimized in a progressive manner using gradient descent by minimizing the $\ell_2$ distance between the produced approximation and the original weights. Finally, as the previous steps were input agnostic, the scaling factors are optimized holistically, calibrating them in student-teacher fashion, with the teacher providing both the data, by autoregressive generation starting from a random token, and the target logits. When

Table 4: **Weights only quantization** results of LLaMA and LLaMA-2 models in terms of perplexity on WikiText2.

| Method | Quantization | LLaMA | | | | LLaMA-2 | | |
|---|---|---|---|---|---|---|---|---|
| | | 7B | 13B | 30B | 70B | 7B | 13B | 70B |
| FP16 | - | 5.68 | 5.09 | 4.10 | 3.53 | 5.12 | 4.57 | 3.12 |
| RTN | | 6.43 | 5.55 | 4.57 | 3.87 | 6.11 | 5.20 | 3.67 |
| GPTQ [16] | | 6.13 | 5.40 | 4.48 | 3.83 | 5.83 | 5.13 | 3.58 |
| AWQ [32] | W4A16 | 6.08 | 5.34 | 4.39 | 3.76 | 6.15 | 5.12 | - |
| OmniQuant [48] | | 5.86 | 5.21 | 4.25 | 3.71 | 5.74 | 5.02 | 3.47 |
| QuIP# [54] | | 5.76 | 5.17 | **4.18** | **3.60** | 5.56 | 4.95 | 3.38 |
| RTN | | 5.96 | 5.25 | 4.23 | 3.67 | 5.72 | 4.98 | 3.46 |
| GPTQ [16] | | 5.85 | 5.20 | 4.23 | 3.65 | 5.61 | 4.98 | 3.42 |
| AWQ [32] | W4A16g128 | 5.81 | 5.20 | 4.21 | 3.62 | 5.62 | 4.97 | - |
| OmniQuant [48] | | 5.77 | 5.17 | 4.19 | 3.62 | 5.58 | 4.95 | 3.40 |
| RTN | | 25.73 | 11.39 | 14.95 | 10.68 | 539.48 | 10.68 | 7.52 |
| GPTQ [16] | | 8.06 | 6.76 | 5.84 | 5.06 | 8.37 | 6.44 | 4.82 |
| AWQ [32] | W3A16 | 11.88 | 7.45 | 10.07 | 5.21 | 24.00 | 10.45 | - |
| OmniQuant [48] | | 6.49 | 5.68 | 4.74 | 4.04 | 6.58 | 5.58 | 3.92 |
| QuIP# [54] | | 5.98 | 5.31 | 4.36 | 3.78 | 5.79 | 5.10 | 3.56 |
| RTN | | 7.01 | 5.88 | 4.87 | 4.24 | 6.66 | 5.51 | 3.97 |
| GPTQ [16] | | 6.55 | 5.62 | 4.80 | 4.17 | 6.29 | 5.42 | 3.85 |
| AWQ [32] | W3A16g128 | 6.46 | 5.51 | 4.63 | 3.99 | 6.24 | 5.32 | - |
| OmniQuant [48] | | 6.15 | 5.44 | 4.56 | 3.94 | 6.03 | 5.28 | 3.78 |
| **QBB (Ours)** | W(4x1)A16g128 | **5.69** | **5.14** | 4.19 | 3.64 | **5.21** | **4.63** | **3.20** |

evaluated across multiple LLM families, our approach matches and outperforms all prior works, setting a new state-of-the-art result using a summation-only based approach.

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
