# OpenReview forum: "QBB: Quantization with Binary Bases for LLMs"
_NeurIPS.cc/2024/Conference — NeurIPS 2024 poster_

### Official Review · Reviewer_waud · 2024-07-08

**Soundness:** 2
**Presentation:** 3
**Contribution:** 2
**Rating:** 3
**Confidence:** 4

**Summary:**

This paper proposes a PTQ technique which decompose the model weitghs into a set of binary matrices. An interactive binarization process and a progressive model distillation procedure are conducted to reduce the quantization error. The paper claims to set a new SOTA of a summation-only based approach.

**Strengths:**

1. This paper works on the problem of quantizing large language model, which is an important research question with practical application and positive social impact.
2. The paper is overall well-written, the method is clearly derived and well-illustrated with figures. The motivation and the methods are easy to follow.
3. Good performance are reported both in terms of preplexity and zero-shot performance on multiple models.

**Weaknesses:**

1. From the novelty prespective, the proposed binary decomposition method is largely similar to previous nonlinear quantization methods line LQ-Net (https://arxiv.org/pdf/1807.10029 ECCV'18). The similarity and difference between the proposed method and LQ-Net should be discussed and the performance should be compared.
2. The paper claims the achieved model can be efficient by removing all the multiplications. However, I don't believe this can be achieved from the formulation of quantized weights in Equ. (1). Although no multiplication is needed to multiply the binary part B with the activation, the multiplication of scaling factor alpha to the activation is still needed, and can be costly when N is large. In fact, the proposed quantization scheme is effectively an N-bit quantization with non-uniform quantization bins, which theortically shouldn't be more efficient than an N-bit linear quantization with integer weights.
3. Deriving from my second point, the efficiency discussion in Sec 4.6 is not concrete. An estimation on exactly how memory and computation is saved comparing to a regular linear W4A16 quantized model should be provided.
4. The comparison is mainly done against PTQ methods with linear weight quantization, yet the proposed method performs non-linear quantization of the weight and requires several epochs of finetuning. This will result in both higher training cost and inference cost comparing to previous methods, making the comparison results unfair.

**Questions:**

1. Please revise the discussion in Sec 4.6 to provide detailed memory and computation savings comparing to regular linear W4A16 quantized models.
2. Please explain in detail about the quantization configuration of the reported W2.3 quantized model

**Limitations:**

The limitation of the work is adequately addressed.

---

> ### Author Rebuttal · Authors · 2024-08-06
>
> We thank the reviewer for their time and comments. We hope to have addressed their remaining concerns below.
>
> **Q1.** _From the novelty prespective, the proposed binary decomposition method is largely similar to previous nonlinear quantization methods line LQ-Net ([https://arxiv.org/pdf/1807.10029](https://arxiv.org/pdf/1807.10029) ECCV'18). The similarities and differences between the proposed method and LQ-Net should be discussed, and the performance should be compared._
>
> **A1.** We disagree, as the similarities are largely superficial. While LQ-Net uses multiple binary bases to emulate n-bit quantization, unlike our approach: they are fully trained in a supervised manner on the task at hand, hence their solution is neither data-free nor is applicable as post-training quantization step; they follow a different formulation, in which the binary basis is found by looking up the quantization interval while the learnable floating basis by solving a linear regression problem. Moreover, no form of distillation, such as the strategy proposed in this work is used.
> In contrast, our process follows a two-step approach, in which we first construct a series of binary bases that closely approximate the original weights, without requiring any activations, on a layer-by-layer basis, directly using gradient descent. We then calibrate the scaling factors using a data-free approach and distillation strategy.
>
> In terms of a numerical comparison, the current LQ-Net formulation would require retraining the model from scratch/finetuning it on specific datasets.
> Thank you for pointing out this work. We will include an ample discussion making the difference clear.
>
> **Q2.** _The paper claims the achieved model can be efficient by removing all the multiplications. However, I don't believe this can be achieved from the formulation of quantized weights in Equ. (1). Although no multiplication is needed to multiply the binary part B with the activation, the multiplication of scaling factor alpha to the activation is still needed and can be costly when N is large. In fact, the proposed quantization scheme is effectively an N-bit quantization with non-uniform quantization bins, which theoretically shouldn't be more efficient than an N-bit linear quantization with integer weights._
>
> **A2.** The reviewer is correct that we don't remove _all_ multiplications, however, we don't claim to remove all multiplications, but _nearly_ all multiplications. The remaining multiplications are not too different from most quantization approaches which will also scale the output using some scalars. Do note however that the 4b-16bit multiplications, occurring for the matmul itself, are removed, and as noted in Sec 4.6, depending on the hardware implementation they can be significantly more energy efficient, requiring fewer physical gates.
>
> **Q3.** _Deriving from my second point, the efficiency discussion in Sec 4.6 is not concrete. An estimation of exactly how memory and computation are saved compared to a regular linear W4A16 quantized model should be provided._
>
> **A3.** Most of the savings are expected to be from the significant reduction in multiplications, converting it to a nearly summation-only approach. This could primarily result in significant energy savings (as pointed out 4-8x). For this aspect we kindly point out to the works cited in Sec. 4.6 [24; 34; 54; 55; 61;] which have a more ample and practical analysis on this, as they specialise on summation-based strategies. In terms of memory savings, on a memory-pressured system, an intuitive direct saving could be by loading one binary basis at a time into memory, overlapping the computation of the first computation with the loading of the data of the second, thus diminishing the peak memory used.
> To obtain exact measurements, an implementation is needed, which as we mentioned in the limitation section, wasn't part of the present work.
>
> **Q4.** _The comparison is mainly done against PTQ methods with linear weight quantization, yet the proposed method performs non-linear quantization of the weight and requires several epochs of finetuning. This will result in both higher training cost and inference cost comparing to previous methods, making the comparison results unfair._
>
> **A4.** There is no additional cost at runtime as the calibration process is performed prior to test time deployment. During calibration, indeed, the process, altogether, depending on the model, requires 4-12hours. As the process is expected to be run once, can be performed on commodity hardware, and requires no data, we consider this time to not be a major concern in practice.
>
>
> **Q5.** _Please explain in detail about the quantization configuration of the reported W2.3 quantized model_
>
> **A5.** The quantization configuration is based on N=4 strategy under the assumption of left-most significant placement (detailed in Sec. 4.6.) We also observe some patterns in terms of quantization errors (see Fig. 6) which should make it possible to adjust the levels further depending on this.

---

> > ### Comment · Reviewer_waud · 2024-08-08
> > **Further discussion on efficiency**
> >
> > I would like to thank the author for the rebuttal. I'm satisfied with the first point.
> >
> > However, I'm still confused on how the proposed method is able to "remove nearly all multiplication" comparing to traditional W4A16 scheme. From my understanding, the linear quantized weights, like those in GPTQ and AWQ, can also be represented as a summation of binary matrices, specifically
> > $$W = \sum_{n=0}^3 2^n B_n,$$
> > with $B_n$ each being a binary matrix. This is a special case of your quantized weight formulation in Eq. (1), with the scaling factor alpha being exponenets of 2. From my understanding, you do not constrain the value of alpha in your formulation. Therefore, I do not see exactly why you can bring additional efficiency comparing to linear weight quantization.
> >
> > From my understanding, the proposed method is more comparable with non-linear weight quantization methods like SqueezeLLM [1], which benefits from memory saving, but not computational efficiency. This makes the reported comparisons with previous linear quantization methods like GPTQ and AWQ unfair.
> >
> > [1] Kim, Sehoon, et al. "Squeezellm: Dense-and-sparse quantization." arXiv preprint arXiv:2306.07629 (2023).

---

> > > ### Author Response · Authors · 2024-08-09
> > > **RE: Further discussion on efficiency**
> > >
> > > We thank the reviewer for considering our rebuttal, and we are glad that the reviewer is satisfied with the first point. We hope to have addressed/clarified your remaining concern below.
> > >
> > > The reviewer is correct that the weights for GPTQ/AWQ can be expressed as a summation of binary matrices too, and we also agree with the parallel drawn with [1] Squeezellm; It is indeed fair to state that under such representation, both approaches will be subject to similar reduction in multiplication (i.e. the remaining multiplications are with the activations, post matrix-multiplication). We will revise the text to make this aspect clear. Do please note, that we don't claim to be more efficient than AWQ/GPTQ, but simply, that depending on the HW, a binary matrices views can be a more efficient framework to operate in. We thank you for this, we will make a parallel to [1] in the paper.
> > >
> > > Regarding the comparisons, we do believe that the numerical comparisons are fair, as our main comparison is with W4, in which case GPTQ/AWQ is not put at a disadvantage.

---

### Official Review · Reviewer_L58J · 2024-07-10

**Soundness:** 3
**Presentation:** 3
**Contribution:** 3
**Rating:** 6
**Confidence:** 4

**Summary:**

This paper introduces Quantization with Binary Bases (QBB), a novel method designed to reduce the computational complexity of large language models (LLMs) by replacing most multiplications with summations. QBB decomposes original weights into a set of binary (1-bit) matrices through an iterative process, optimizing them to minimize quantization error. Additionally, the quantized model is refined via Knowledge Distillation. Tested across multiple LLM families, QBB demonstrates superior performance compared to existing methods.

**Strengths:**

1. This method's utilization of binary matrices and scaling vectors to approximate original weights marks a significant advancement, effectively reducing computational complexity.
2. Combining post-training quantization with knowledge distillation is a commendable approach. The performance has notably improved through the application of Knowledge Distillation after quantization.
3. The paper offers extensive experimental results that showcase the effectiveness of QBB across diverse LLMs and datasets.

**Weaknesses:**

1. I believe comparing it with non-uniform quantization methods, such as QuIP#, would enhance this paper, as it does not utilize uniform quantization.
2. Regarding progressive weights quantization, I'm unclear whether starting with a quantized model means that $W$ in Equation 4 represents quantized weights rather than full-precision weights. If so, this approach may not be optimal for the quantization step.
3. For Training details, does each layer need 15000 iterations? Can you provide the details of runtime and overhead?
4. How were the values of $s_1$ and $s_2$ specified in Equation 7?

**Questions:**

Please refer to Weakness.

**Limitations:**

Please refer to Weakness.

---

> ### Author Rebuttal · Authors · 2024-08-06
>
> We thank the reviewer for their time and comments. We hope to have addressed their remaining concerns below.
>
> **Q1.** _I believe comparing it with non-uniform quantization methods, such as QuIP#, would enhance this paper, as it does not utilize uniform quantization._
>
> **A1.** Thank you for your suggestion. We already report results for QuIP# in Table 3. We will expand this to include more configurations, with different levels of quantization, making this aspect also more clear.
>
> **Q2.** _Regarding progressive weights quantization, I'm unclear whether starting with a quantized model means that_
>
> **A2.** By this, we mean that instead of directly binarizing the full/half precision weights, we start the process from a model quantized using an off-the-shelf 4-bit quantization method. This reduces the initial informational gap and is well aligned with the idea of gradually dropping the number of bits used in literature. Please see also Table 4.
>
> **Q3.** _in Equation 4 represents quantized weights rather than full-precision weights. If so, this approach may not be optimal for the quantization step._
>
> **A3.** In eq. 4, W represents the to-be-approximates weights, they can be full precision or quantized to a higher number of bits. B_i represents the constructed binary matrices. As num_bits(W) is always higher than num_bits(B) this is generally not a concern. Moreover, this steps only corrects the remaining error that is incurred after the first step.
>
> **Q4.** _For Training details, does each layer need 15000 iterations? Can you provide the details of runtime and overhead?_
>
> **A4.** Yes, this is performed on a layer-by-layer basis, we could likely reduce this significantly using an early stopping scheduler, but for simplicity, we opted for the same number of iterations in all cases. In terms of overhead, this takes a few hours (4-12) on a A100, depending on the model size. However, as the process is only run once, there is no additional overhead at test/runtime.
>
> **Q5.** _How were the values of and specified in Equation 7?_
>
> They were selected empirically using one of the model variants (i.e. the smallest), and then used for the rest of the models too.

---

### Official Review · Reviewer_XBMG · 2024-07-12

**Soundness:** 3
**Presentation:** 3
**Contribution:** 3
**Rating:** 5
**Confidence:** 5

**Summary:**

This paper proposed QBB by discomposing the original weights into 4 binary matrics and scaling factor.
To compensate the error, two techniques are further proposed:
1. use gradient descent to find the optimal set of binary matrices and scaling vectors
2. use knowledge distillation to optimize the scaling vector only.

**Strengths:**

- This paper present a thorough analysis, including detailed ablation studies and analysis of different components.
- This paper has clear presentation: Well-organized, with effective use of figures and tables to support the content.

**Weaknesses:**

- The technique of decomposing the weights into several binary matrics is similar to BiLLM [1]. In BiLLM, they proposed residual approximation method, which is just the same to QBB when N=2. And this paper was published on arxiv on Feb. 2024.

- In the related work, QBB listed the PB-LLM, which is a mixed-precision framework that achieved around 2-bit quantization. However, the experiments in Table3 did not involve it. Additionally, BitDistiller [2] also conducted experiment on 2-bit weight quantization.

- This paper did not provide the code in supplementary material.

- In QBB, two additional adjustments were conducted to minimize the error, which involves additional training cost. However, I haven't found any experiments w.r.t. training cost. Correct me if I am wrong.

- From my understanding, the computational cost of QBB is between PTQ and QAT. You should compare with PTQ and QAT method separately. For QAT, LLM-QAT and Bitdistiller should be involved.

- From line 154 to 157, you proposed progressive weights quantization technique by first quantizing weights to 4-bit and then applying the QBB. There is no additional process between these two procedure. From my point, I think it is useless and uneffective. There is no experiment that can prove that your progressive weights quantization is better than directly quantizing.

- From Figure 6, when we set the N to 5, it is more stable than 4. Why not just use 5?

- In section 3.2, "Data-free" is not properly used. Because I find that you utilized the generated data by LLM. It is not completely data-free. Please refer to [3].

- Teacher-student swaps can easily cause training collapses, especially for floating-point & low-bit models. I am more curious about any strategies employed  to alleviate this problem.

[1] https://arxiv.org/pdf/2402.04291
[2] https://arxiv.org/abs/2402.10631
[3] https://arxiv.org/abs/1906.04721

**Questions:**

See weakness.

**Limitations:**

This paper missed some important reference. The experiments part should be put more efforts.

---

> ### Author Rebuttal · Authors · 2024-08-06
>
> We thank the reviewer for their time and comments. We hope to have addressed their remaining concerns below.
>
> **Q1.** _On: The technique of decomposing the weights into several binary matrics is similar to BiLLM [1], published on arxiv on Feb. 2024._
>
> **A1.** Thank you for pointing out [1], we weren't aware of it. Kindly note that [1] was not published in a peer-reviewed venue at the time of the NeurIPS deadline. Nevertheless, we will cite and discuss it in our updated manuscript.
>
> Regarding, the differences with [1]: [1] focuses on applying different quantization strategies for the salient and non-salient weights, identified using a structural search based on the Hessian Matrix. The salient columns then use a secondary residual binarization (i.e. to better preserve the information in the salient channels). The remaining non-uniformly distributed weights are split intro two groups by searching the optimal split. The work aims generally to exploit the sparsity of information present in the weights, allocating the bits accordingly and focusing on extreme quantization levels.
>
> In contrast, our approach directly approximates the weights using a sum of binary matrices, that are learned on a layer-by-layer basis, without using any calibration data (as opposed to [1]), by gradient descent). Each new binary weight is trained in a cascaded manner. Our approach generalizes to different numbers of binary matrices, doesn't make use of any splitting algorithm, nor aims to identify/tie the binarization process to the weights' saliency.
>
> Thereafter, unlike [1]  we follow a data-free calibration of the scaling factors using self-generated sequence that starts from randomly sampled tokens. Finally, the calibration process is placed within a newly adapted distillation strategy.
>
> We believe that these are significant methodological differences that distant our work from [1] (an arxiv paper).
>
>  **Q2.** _On comparison with PB-LLM & BitDistiller[2] that achieve 2-bit W quantization._
>
> **A2.** As we didn't focus on 2-bits or less, we haven't included such methods in the Tables. However, following your suggestion, we will include [2] in the result tables (i.e. 5.97[2] vs 5.21 (this work) on a LLamA-7B-2), noting that the results won't be directly comparable.
>
> **Q3.** _This paper did not provide the code in supplementary material._
>
> **A3.** Due to internal processes and timelines, unfortunately this wasn't possible to do at the time of submission.
>
> **Q4.** _In QBB, two additional adjustments were conducted to minimize the error, which involves additional training cost. However, I haven't found any experiments w.r.t. training cost._
>
> **A4.** Depending on the model choice, the training take 4-12 hours on A100 GPU. This can likely be reduced by early stopping, as in many cases the process converges long before the end of the scheduler. However, for simplicity, we kept the same value.
>
> **Q5.** _The computational cost of QBB is between PTQ and QAT. You should compare with PTQ and QAT: LLM-QAT and Bitdistiller should be involved._
>
> **A5.** We haven't included a comparison with the two works mentioned as our work only performs a calibration process using a data-free approach, unlike LLM-QAT which performs a full model finetuning. E.g. LLM-QAT requires 8 A100 GPUs at a batch size = 1 per GPU to perform full model finetuning. In comparison, our approach can be run on a single GPU.  Nevertheless, we see the potential usefulness of such comparisons, and following your request, we will include them in the updated manuscript.
>
> **Q6.** _From L154-157, you proposed progressive weights quantization technique by first quantizing weights to 4-bit and then applying the QBB. There is no additional process between this two procedure. From my point, I think it is useless and uneffective. On no experiment to show this._
>
> **A6.** We ablate the impact of different initialization strategies in Table 4, where we showcase both the importance of starting from 4-bit weights and the method used to construct the 4-bit weights. The results show this step to be necessary. The intuition behind this is that the gap between the sum(binary weights) and fp16 ones are larger, making the optimisation process more challenging.
>
> **Q7.** _From Figure 6, when we set the N to 5, it is more stable than 4. Why not just use 5?_
>
> **A7.** We didn't as the additional cost didn't justify to us the additional costs (see Fig. 6).
>
> **Q8.** _In section 3.2, "Data-free" is not properly used. Because I find that you utilized the generated data by LLM. It is not completely data-free. Please refer to [3]._
>
> **A8.** We kindly disagree: the data-free assumption is not violated. The sequences used for training are produced by the model itself, starting from a random token. There is no real (i.e. collected) data used in the process, nor techniques based on manual intervention (e.g. prompting). As the starting token is random, this process is more akin to self-distillation on random inputs.
> We note that such discriminative models, such as those used in [3] (i.e. MobileNet archs) are not comparable to generative models such as LLMs, as they are from different families of models, and the former are unable to produce sequences of data given a random sample. Hence, this is not a violation of the setting, but simply a difference between the capability of vision models vs an LLM when exposed to randomly constructed points.
>
> **Q9.** _Teacher-student swaps can easily cause training collapses, especially for floating-point & low-bit models. I am more curious about any strategies employed to alleviate this problem._
>
> **A9.** In our case, the first stage aims to independently (i.e. on a layer-by-layer basis) approximate the weights of each layer, using a set of binary weights. Post approximation, these layers are expected to already behave similarly with the target weights. Thanks to this, the discrepancies between such layers is small, making the swapping stable in general.

---

### Official Review · Reviewer_vSy1 · 2024-07-12

**Soundness:** 3
**Presentation:** 3
**Contribution:** 2
**Rating:** 6
**Confidence:** 4

**Summary:**

This research brings the sum of binary bases to the post-training quantization of the large language models. The authors propose a three-step algorithm. In step1: taking the sign of the full-precision weights and use the norm of the weights as the scalings, step2: adjusting the binary bases using gradient descent, step 3: adjusting scales using teacher-student (knowledge distillation).

Step 1 was initiated in computer vision [49], and step 2 introduced in [36]. Set of binary bases for inference was suggested in [36] and later in LQ-Net. https://arxiv.org/pdf/1807.10029

The full binary base for efficient inference is not novel, authors tried to adapt [36] for post-training of large language models, which necessitates step 3 for a better performance. The paper needs to clarify its connection with [36] and LQNet. I was confused in the middle of reading the paper about the main contribution of the paper. My final judgment is "this paper is an adaptation of existing ideas," but I congratulate the authors on putting considerable effort into bringing it to post-training and large language model context.

**Strengths:**

The paper proposes step 3 to make the performance close to the state-of-the-art post-training such as omi-quant, GPTQ, AWQ.

**Weaknesses:**

I doubt that Table 3 fairly shows the strength and the weakness of the method. First explain what W4A16g128 mean. I suppose it means weight quantized to 4 bit fixed point, activation to floating point 16 (half precision or brain float?) or maybe fixed point, g128 referees to the grouping size of the weight quantization for the fixed point representation.
- SpQR baseline is missing in Table 3. https://arxiv.org/abs/2306.03078

**Questions:**

- Did you try your method on the fixed-point activations? if yes please specify, if not please make it clear in the paper.
- Why competing methods are on 4 and 3 bits but yours is 2.3? Is it a mistake or 2.3 is the average bit to be 2<2.3<3. I think it is fair to report your result on W4 and W3 just like the other baselines and then show it on a lower lower average bit.

**Limitations:**

The paper focuses on LLamA models. I would prefer a similar comparison for other mainstream language models as well.

---

> ### Author Rebuttal · Authors · 2024-08-06
>
> We thank the reviewer for their time and comments. We hope to have addressed their remaining concerns below.
>
> **Q1.** _"On doubts that Table 3 fairly shows the strength&weakness of the method.  First explain what W4A16g128 mean._
>
> **A1.** We believe that the comparison is fair, as we align our setting to that of prior methods, e.g. AWQ. Regarding the meaning of W4A16g128, you are correct. The example denotes a quantization scheme that makes use of 4-bit fixed point quantization for the weights, with a grouping size of 128 and half-precision (i.e. fp16) activations. We will detail this in the updated manuscript.
>
> **Q2.**  _SpQR baseline is missing in Table 3._
>
> **A2.** As suggested, we will include the baseline in Table 3, thank you! We note that the conclusions don't change (i.e.: LLaMA-7B @ pp on Wiki2, 5.69 (Ours) vs 5.87 (SPQR));
>
> **Q3.** _Did you try your method on the fixed-point activations? if yes please specify, if not please make it clear in the paper._
>
> **A3.** We performed an ablation in Table 5. Our approach only affects the weights, hence it's largely agnostic to the quantization strategy used for the activations. We will make this clearer.
>
> **Q4.** _On why no results with W4 and W3 just like the other baselines and then show it on a lower average bit._
>
> **A4.** Performance doesn't improve significantly beyond this to justify the costs, this is a consequence of initializing our model from W4 quantized weights.
>
> **Q5.** The paper focuses on LLamA models. I would prefer a similar comparison for other mainstream language models as well.
>
> **A5.** We mainly focused on LLamA (v1 and v2) models as this facilitated the comparison with prior work. We also already include results for the Phi-2 model (Table 1), and in addition, we tested it on Mistral-7B where we saw similar results (i.e. +0.1 perplexity improvement compared with the direct GPTQ baseline)

---

### Decision · Program_Chairs · 2024-09-25

**Decision:**

Accept (poster)

**Comment:**

This paper presents a novel approach called "Quantization with Binary Bases (QBB)" for low-bit quantization of large language models (LLMs). The method decomposes original weights into binary matrices, significantly reducing computational complexity by replacing most multiplications with summations. The authors demonstrate that QBB achieves state-of-the-art performance across various LLMs.

Reviewers acknowledged the technical soundness of the approach but raised concerns about its novelty, particularly in relation to existing methods like LQ-Net. They also questioned the paper's claims regarding computational efficiency and the fairness of comparisons with linear quantization methods. Some reviewers suggested additional experiments and comparisons to strengthen the paper's contributions.

In response, the authors provided detailed clarifications and additional comparisons, addressing the reviewers' concerns. Most reviewers agreed that the paper offers a solid contribution to the field, particularly in its innovative application of binary matrices to LLM quantization. Considering the paper's strengths, including its thorough experimentation and practical implications, the AC recommends acceptance.